# Effects of Iron Powder Addition and Thermal Hydrolysis on Methane Production and the Archaeal Community during the Anaerobic Digestion of Sludge

**DOI:** 10.3390/ijerph19084470

**Published:** 2022-04-08

**Authors:** Xiuqin Cao, Yibin Wang, Ting Liu

**Affiliations:** 1School of Environment and Energy Engineering, Beijing University of Civil Engineering and Architecture, Beijing 100044, China; 201403020120@stu.bucea.edu.cn (Y.W.); 18811584886@163.com (T.L.); 2Key Laboratory of Urban Stormwater System and Water Environment, Ministry of Education, Beijing University of Civil Engineering and Architecture, Beijing 100044, China

**Keywords:** anaerobic digestion, biogas, archaeal community, hydrogenotrophic methanogens, acetoclastic methanogens

## Abstract

The conventional anaerobic digestion of sludge has the disadvantages of long digestion time and low methane production. Pretreatment is often used to mitigate these problems. In this study, three pretreatment methods, namely, the addition of iron powder, high-temperature thermal hydrolysis, and a combination of these methods, were compared for application with conventional continuous anaerobic digestion reactors. The results showed that pretreatment improved methane yield by 18.2–22.9%, compared to the control reactor (conventional anaerobic digestion). Moreover, it was recognized that the archaeal community in the sludge underwent significant changes after pretreatment. Specifically, the addition of iron powder reduced the diversity in the archaeal community, but increased the abundance of hydrogenotrophic methanogens without changing the community composition. Thermal hydrolysis at high temperatures had the reverse effect, as it increased the diversity of the archaeal community but inhibited the growth of acetoclastic methanogens. In the case of the combined pretreatment, the thermal hydrolysis had a dominant influence on the archaeal community. By comparing the changes in functional gene content, it was found that the functional abundance of the archaeal community in the transport and metabolism of carbohydrates, lipids, and amino acids was higher after pretreatment than in the control group.

## 1. Introduction

As an energy-consuming process, sewage treatment processes involve the use of energy to dissipate energy during operation. As China heads for carbon neutrality, the realization of carbon neutralization in sewage treatment plants is very important. As a source of renewable energy, methane is not only an important product of the anaerobic digestion process, but also the key to carbon neutrality in sewage plants [1,2,3]. Conventional sludge treatment based on anaerobic digestion includes the following four steps: hydrolysis, acidogenesis, acetogenesis, and methanogenesis [4]. However, it is a slow process with several disadvantages, such as large digester volume requirements and long residence time [5]. This is because hydrolysis, the first step of anaerobic digestion, is essentially the rate limiting step of the process. Therefore, the anaerobic digestion process can be improved by pretreatment to increase the yield of methane [6,7]. At present, the commonly applied methods of sludge pretreatment include mechanical, microwave, thermal, electrical, and biological methods and their combinations [8,9,10,11,12,13].

Among these pretreatment methods, THP (thermal hydrolysis pretreatment) is considered the most reliable and is widely used. THP can hydrolyze large particulate organic matter in sludge into small particles using high temperatures and high pressure [14]. At the same time, it can also rupture cells and promote the outflow of intracellular substances to improve methane production [15,16]. At present, the temperature range used in thermal hydrolysis research is 40~180 °C. Based on the operating temperature range, the process can be divided into low-temperature (temperature lower than 100 °C) and high-temperature thermal hydrolysis (temperature higher than 100 °C) [17]. Some studies show that the optimal temperature of low-temperature thermal hydrolysis is 90 °C [18], and that for high-temperature thermal hydrolysis is 160–170 °C [5,19,20], but it is generally believed that high-temperature thermal hydrolysis performs better than low-temperature thermal hydrolysis. Xue et al. compared low-temperature and high-temperature thermolysis in sludge treatment and showed that the biogas output increased by 8.8–12.3% after low-temperature thermolysis, and by approximately 16.8–36.4% after high-temperature thermolysis [21]. Liao et al. used low-temperature thermal hydrolysis (60, 70, 80 °C) to treat sludge with high solid content and found that the cumulative methane production increased by 7.3–24.4% compared with the untreated group [22]. Bougrier et al. used thermal hydrolysis at 135–190 °C to treat the sludge, so that the maximum increase in methane in anaerobic digestion reached 25% [23].

In recent years, the addition of iron powder has also been tried as a pretreatment method. The addition of iron powder can promote the decomposition and conversion of propionic acid and the formation of acetic acid, and improve the volatile fatty acid (VFA) yield in the process of hydrolysis and acidification [24]. Hao et al. comprehensively summarized the decrease in the oxidation-reduction potential (ORP) of iron during the anaerobic digestion process, its impact on the physiological and biochemical characteristics of anaerobic microorganisms, and the impact of hydrogen evolution from iron corrosion on microbial enzyme activities and the methanogenic process [25]. Niu et al. found that when the iron powder dosage was 31.13 g·kg^−1^TS (18.8 g·kg^−1^VS), the effect of anaerobic digestion of sludge was the best, and the methane yield increased by 21.29% [26]. Wei et al. added 1 g·L^−1^, 4 g·L^−1^, and 20 g·L^−1^ of iron with zero valency to the anaerobic digestion system. The results showed that the cumulative methane yield was the largest when 4 g·L^−1^ of zero valent iron was added, and the methane yield increased by 26.9% [27]. Zhang et al. added sulfidated nanoscale zero-valent iron (S-nZVI) into anaerobic digestion systems of food waste under ammonia stress. This practice improved the methane production and enriched the species related to ammonia-tolerant hydrogenotrophic methanogenesis [28]. Lim et al. added nano-zerovalent iron into the reactor; this promoted the generation of VFA but inhibited methanogens at the beginning of the reaction [29].

Thermal hydrolysis pretreatment enhances the anaerobic digestion methane production process from the hydrolysis stage. The addition of iron powder strengthens the acidogenesis, acetogenesis, and methanogenesis stages. Therefore, combining the two pretreatment methods to strengthen the anaerobic digestion methane production process in four stages has been considered. However, research combining these two pretreatment methods to treat sludge has not previously been undertaken. Therefore, the purpose of this experiment was to compare the combined treatment method with the single treatment methodology and to determine the optimal method to increase methane production. The effects of different pretreatment methods on the archaeal community in anaerobic digestion were also investigated, and the reasons for these changes were explored. For this, sludge samples were pretreated using three different methods, namely, 30 mg·g^−1^VS iron powder, high-temperature thermal hydrolysis at 170 °C, and a combination of the two methods to improve methane production.

## 2. Materials and Methods

### 2.1. Inoculum and Feedstock

The dewatered sludge and anaerobically digested sludge used in this study were taken from a sewage treatment plant in Beijing. Dewatered sludge was stored in a refrigerator at 4 °C before use, and anaerobically digested sludge was used immediately after being taken from the anaerobic digestion tank. The sludge used in the high-temperature thermal hydrolysis experiment was from a mixture of the dewatered sludge and the anaerobic-digested sludge. The mixture underwent high-temperature thermal hydrolysis in a thermal reactor, and the uniform heat transfer of the sludge was ensured by mechanical stirring in the reactor.

### 2.2. Determination of Physiochemical Index

The pH was measured using a pH meter (Mettler Toledo S210, Mettler Toledo, Zurich, Switzerland). TS (total solids) and VS (volatile or organic solids) were determined by the gravimetric method. The sludge was dried in an oven at 105 °C for 24 h, placed in a muffle furnace and burned at 550 °C for 2 h, and the corresponding weight loss was used to calculate the TS and VS [30]. We used NaOH solution to absorb CO_2_ in gas production and the content of methane was calculated according to the volume of NaOH solution discharged. The sludge was centrifuged at 10,000 r/min for 20 min. The supernatant was filtered through a 0.45 μm mixed-fiber membrane, and the filtrate was used for the subsequent measurement of dissolved organic matter. The SCOD (Soluble Chemical Oxygen Demand) was measured using a HACH COD rapid analyzer, DRB200 digester, and DR6000 spectrophotometer; the alkalinity (ALK) content was measured using acid-based indicator titration; the TAN (Total Ammonium Nitrogen) content was determined by the nanoreagent method; and the VFA (Volatile Fatty Acids) content was quantified using colorimetry [18]. HACH DR6000 spectrophotometers were used for the measurements. The basic characteristics of the sludge are shown in Table 1 and Table 2.

### 2.3. Reactor Setup and Experimental Design

The total volume of the reactor used in the current study was 2 L, and the working volume was 1.4 L. The reactor was wrapped with a heating belt and insulating material, and was equipped with a temperature-sensing device to control the temperature at 37 ± 1 °C. It used mechanical stirring: the stirring speed was 120 r/min, and the sludge was stirred twice a day for 1 h each time. The methane that was produced was collected and recorded every day. There were two variables in the experiment, namely, iron powder addition and high-temperature thermal hydrolysis. Studies have shown that the optimal temperature range for high-temperature thermal hydrolysis is 160–170 °C [23], and so, the temperature was set to 170 °C. When Suanon et al. added iron powder to anaerobic digestion sludge, the cumulative methane yield increased by 40.8%, and the dosage of iron powder was 33 mg·g^−1^VS [31]. Based on this, the iron powder dosage was set as 30 mg·g^−1^VS.

### 2.4. Microbiological Analysis

To clarify the effects of iron powder addition and high-temperature thermal hydrolysis (HTHP) on the sludge, the sludge samples were placed in four reactors. After 21 days of the reaction, the sludge in the four reactors was taken to analyze the metagenomic sequencing of the archaeal community. The samples were first pretreated, and the DNA was extracted for PCR amplification. The V3-V4 hypervariable regions of the 16S rDNA genes of the archaeal community were amplified by the primers M-340F (CCCTAYGGGGYGCASCAG) and GU1ST-1000 R (GGCCATGCACYWCYGTCTC). Subsequently, the primers 349F (CCCTACACGACGCTCTTCCGATCTN (barcode) GYGCASCAGKCGMGAAW) and 806R (GACTGGAGTTCCTTGGCACCCGAGAATTCCAGGACTACVSGGGTATCTAAT) were used to amplify the first PCR products. After the metagenomic sequencing, the ribosomal database project (RDP) algorithm was used to define the operational taxonomic units (OTUs) at a 97% sequence similarity threshold. Finally, according to the number of OTUs, the α diversity and β diversity of the archaeal were analyzed.

### 2.5. Calculations and Statistics

Origin software (2019b, OriginLab, Massachusetts, USA) and IBM SPSS Statistics (25.0, IBM, New York, NY, USA) were used for data processing and statistical analysis.

## 3. Results

### 3.1. Performance of Anaerobic Digestion Reactors

#### 3.1.1. Methane Production Efficiency

By the 21st day, anaerobic digestion was completed in all the reactors. The total methane yields of the reactors R_ctrl_ (the control group), R_iron_ (addition of iron powder), R_high_ (high thermal hydrolysis pretreatment), and R_mix_ (combination of two pretreatment methods) were 87.15, 102.97, 107.10, and 106.22 mL·g^−1^VS (Figure 1). Compared with the R_ctrl_, the values of R_iron_, R_high_, and R_mix_ increased by 18.2%, 22.9%, and 21.9%, respectively. This shows that the addition of iron powder and the use of high-temperature thermal hydrolysis can promote an increase in methane production. This result is consistent with that of previous studies. Some studies have shown that methane production increases by approximately 16.8–36.4% after high-temperature thermal hydrolysis is performed [21].

Compared with R_ctrl_, the daily methane production efficiency of R_iron_ was higher; R_ctrl_ stopped producing methane on the 14th day and R_iron_ completed the anaerobic digestion process on the 13th day. At the first peak of methane production, the daily methane production of R_iron_ was about twice that of R_ctrl_, and reached 1.4 times at the second peak of methane production. The daily methane yield of R_high_ and Riron was better than R_ctrl_. The R_high_ completed the anaerobic digestion process in 17 days, which was three days more than R_ctrl_. Like R_ctrl_, R_iron_ showed the first methane production peak on the second day, and the methane production trend was similar to R_ctrl_, followed by two peaks on the 4th and 13th days. This may be because HTHP can promote the dissolution and liquefaction of the macromolecular organic particles, release a large number of organic substances such as protein and polysaccharide, and promote the decomposition of macromolecular substances into substances with smaller molecules, which is conducive to improving the bioavailability of the anaerobically digested sludge and increasing methane production [32].

The first methane production peak of R_mix_ occurred on the 7th day, which showed a relative lag compared with other reactors, and methane production was completed on the 21st day. This may be because the two pretreatment methods acted on microorganisms at the beginning of the reaction, resulting in the maladaptation of the microorganisms to the reactor environment; however, several production peaks followed, and the total methane production was significantly higher than that in R_ctrl_ indicating the presence of high-efficiency archaeal communities in the reactor. This improves the production efficiency of the methane. The methane production process in R_mix_ was similar to that in R_high_, although the daily methane production efficiency fluctuated more than that of R_ctrl_ and R_iron_. The modified Gompertz model (Equation (1)) is a validated empirical nonlinear regression model that is often used to simulate the methane accumulation curve [14]:(1)$$G\left(t\right)=G_{0}\times \exp\left\{-\exp\left[\frac{R_{\max}\times e}{G_{0}}(\lambda -t)+1\right]\right\}$$
where *G*(t) is the cumulative methane production, *G*_0_ is the maximum methane production, *λ* is the lag phase, t is time, and *e* is the exp(1) = 2.71828. R_m__ax_ represents the maximum production rate of methane. Compared with R_ctrl_, there was an increase of 43.6% in R_iron_, indicating that the addition of iron powder can promote the growth of methanogens. The values of R_high_ and R_mix_ were lower than R_ctrl_, which may be because of the maladaptation of the methanogens to the environment in the reactors after HTHP, resulting in the reduction in methanogen growth rate. The value of R_mix_ is higher than R_high_, which indicates that the growth of methanogens is determined by the two pretreatment methods. However, the total quantity of methane production is lower than R_high_. It is necessary to continue to adjust the quantity of iron powder under high-temperature thermal hydrolysis at 170 °C.

#### 3.1.2. VFA Accumulation and pH Fluctuation

In general, the accumulation of VFA in all the reactors matched the variations in the methane production (Figure 2). The first methane production peak in R_ctrl_, R_iron_, and R_high_ occurred one day before the cumulative peak of VFA, and the first methane production peak in R_mix_ occurred the day after the cumulative VFA peak. At the same time, the changing trend of VFA in all the reactors was roughly the same, first increasing and then decreasing gradually. Specifically, in R_ctrl_ without pretreatment, the VFA level was 1101.30 mg·L^−1^ at the beginning of the reaction. After pretreatment, the initial VFA levels of R_high_ and R_mix_ reached 1586.76 mg·L^−1^. Studies have shown that when the VFA levels in the reactor exceeds 1500.00 mg·L^−1^, the operation of the reactor will be affected [33]. At the beginning of the experiment, the VFA levels of R_ctrl_ increased rapidly and reached the highest value of 2867.30 mg·L^−^^1^ the next day. Subsequently, it decreased gradually until 1432.52 mg·L^−1^ at the end of the reaction. In contrast, the VFA levels of R_iron_ reached a peak of 3393.93 mg·l^−1^ the next day, which was 1.2 times that of R_ctrl_. One study showed that the addition of iron powder can promote an abundance of hydrogen- and acetic acid-producing bacteria, resulting in an increase in VFA levels, until the end of the experiment when the VFA levels decreased to 1564.64 mg·L^−1^, indicating that the biological enhancement effect of adding iron powder promoted the growth of acetoclastic methanogens and consumed the VFA accumulated in the reactor [34].

The VFA levels of R_high_ and R_mix_ reached 4885.73 mg·L^−1^ and 5277.08 mg·L^−1^ on the second and fourth days, respectively, corresponding to 1.7 and 1.8 times the value of R_ctrl_. The excess VFA levels may be the reason why the methane production in R_high_ and R_mix_ was lower than that in R_ctrl_. Thermal hydrolysis promotes the dissolution and release of organic matter and also provides sufficient substrate for the formation of VFA. Since the two pretreatment methods acted simultaneously in R_mix_, the VFA level was also the highest among the four reactors. At the end of the reaction, the VFA levels of R_high_ and R_mix_ decreased to 2111.83 mg·L^−1^ and 1471.58 mg·L^−1^. Thus, rapid VFA consumption was observed in R_high_ and R_mix_ compared to R_ctrl_. This is because the biological function of the archaeal community screened by the environment in the reactor was strengthened, and the accumulation of VFA was finally eliminated. At the same time, the VFA level of R_mix_ was lower than R_high_, indicating that iron powder can strengthen the utilization of acetic acid by archaeal. Overall, by the end of the experiment, the VFA levels of R_iron_, R_high_, and R_mix_ were lowered by 8.92–27.13% more than R_ctrl_. At the same time, with the increase in VFA production, the substrate availability was sufficient for the archaeal community, and the methane production reached a peak. Then, the generation efficiency of acetic acid became lower than the consumption efficiency, resulting in a continuous decline in VFA levels.

The changing trends of pH in all the reactors were the same; pH continued to rise from the beginning of the reaction (Figure 3), but was maintained in the optimal range of anaerobic digestion (6.5–8.5) [35]. This may be because ammonia is an important nitrogen source for the pH-stabilizing agent in the neutralization of VFA. At the beginning of the experiment, with the increase in VFA levels, the ammonia levels also continued to rise and stabilize. At the same time, when iron powder was added into the anaerobic digestion reactor, it reacted with the water and organic matter, resulting in an increase in the pH. This shows that the effect of iron powder and ammonia nitrogen on the pH were greater than VFA, and the anaerobic digestion system itself had a regulating effect. So, the pH was relatively stable in the anaerobic digestion process.

### 3.2. Microbial Community Composition and Dynamics

#### 3.2.1. Richness and Diversity Analysis of Archaeal Community

A total of 1534, 1493, 1648, and 1848 OTUs from R_ctrl_, R_iron_, R_high_, and R_mix_, respectively, were obtained based on a sequence similarity of 97%. This indicates that the addition of iron powder reduced the OTU number of the archaeal communities, the effect of high-temperature thermal hydrolysis was opposite, and the effect was better than the addition of iron powder. A Venn diagram showed that R_ctrl_ and R_iron_ were most similar, sharing 219 OTUs. This was followed by R_high_ and R_mix_ that shared 197 OTUs, and then R_iron_ and R_high_ that shared 194 OTUs, R_iron_ and R_mix_ that shared 176 OTUs, R_ctrl_ and R_high_ that shared 169 OTUs, and R_ctrl_ and R_mix_ that shared 155 OTUs. Meanwhile, 76 OTUs were shared by all four samples (Figure 4). This shows that, after different pretreatments, the archaeal communities changed greatly, and the mixed treatment had the greatest impact.

In this experiment, the alpha diversity of the archaeal communities (based on the Shannon index), in descending order, followed the pattern: R_ctrl_, R_iron_, R_mix_, R_high_ (Figure 5). This shows that the added iron powder and high-temperature thermal hydrolysis resulted in the formation of a specialized community throughout the experiment, which resulted in a more efficient AD process. This can also be seen from the methane production, where R_iron_, R_mix_, and R_high_ all had more methane production than R_ctrl_. At the same time, the alpha diversity of archaeal in R_iron_ was less than R_high_. This indicates that high-temperature thermal hydrolysis can result in a more specialized community than the addition of iron powder. However, the archaeal richness (based on the Ace index) followed the pattern R_high_, R_mix_, R_ctrl_, R_iron_ (Figure 5), in descending order, hinting that high-temperature thermal hydrolysis was conducive to improving archaeal richness during the AD process, while the added iron powder had the opposite effect.

Beta diversity based on the principal component analysis showed a clear archaeal community dynamicity between the different conditions of the experiment (Figure 6). Samples from R_ctrl_ and R_iron_ tended to cluster together, suggesting that R_ctrl_ and R_iron_ shared similar methanogen communities; that is, the addition of iron powder resulted in the smallest change to the archaeal community. The archaeal communities of R_mix_ and R_high_ were more similar, showing that high-temperature thermal hydrolysis changed the archaeal community more than the addition of iron powder, and played a dominant role among the two pretreatment methods.

#### 3.2.2. Methanogenic Archaeal Community for Different Pretreatments

At the phylum level (Figure 7), the Euryarchaeota phylum constituted the largest part of the considered samples of the Archaeal domain with an average percentage of 99.82%. Methanomicrobia was the most abundant class (77.58–92.95%) among Euryarchaeota with an average value of 85.59% across all samples. Thermoplasmata and Methanobacteria classes were also detected. Within Euryarchaeota, the presence of seven archaeal orders was established. Methanosarcinales (average 45.25%) and Methanomicrobiales (average 40.32%) were the most dominant. This was followed by Methanomassiliicoccales (12.29%) and Methanobacteriales (1.93%). Small fractions of Nitrososphaerales and Methanocellales were also detected (less than 1%). Among the eleven families identified, Methanomicrobiaceae (38.61%), Methanosarcinaceae (33.73%), Methanomassiliicoccaceae (12.29%), Methanotrichaceae (11.51%), and Methanobacteriaceae (1.93%) were dominant.

The methanogens detected at the genus level (Figure 8) included *Methanoculleus*, *Methanomassiliicoccus*, *Methanobacterium*, and *Methanobrevibacter*; those detected in this study were hydrogenotrophic methanogens. Their proportion in all the reactors reached 44–63%, highlighting the importance of hydrogenotrophic methanogens in anaerobic digestion. The most notable among these was *Methanoculleus*, which was the dominant methanogen in the four reactors, indicating that it occupies an important position in the methane production process. It uses hydrogen and formate or carbon dioxide as a substrate to synthesize methane, and belongs to the order *Methanomicrobiales*. Previous studies have reported that it can tolerate high salt and high ammonia nitrogen conditions, and has an important role in pH and VFA fluctuations [36]. The relative abundance in R_ctrl_ was 30.85%, which increased by 10.51% and 6.51% in R_mix_ and R_high_, respectively, and increased by 14.16% in R_iron_. This indicates that pretreatments strengthened its position in the dominant genus. *Methanoculleus* can use ethanol and some secondary alcohols as electron donors, as they produce methane [37]. This is also one of the important reasons for increasing methane production.

However, the relative abundance of *Methanobacterium* as a hydrogenotrophic methanogen was reduced. At the same time, an interesting observation was that the OTUs of *Methanothrix*, which indicated the relative abundance of *Methanothrix* in R_iron_, were reduced by 7.48%, while remaining undetected in R_mix_ and R_high_. This shows that the three pretreatments had differing inhibitory effects on growth. This shows that the addition of iron powder and thermal hydrolysis can eliminate the uniformity of the archaeal community and make it more efficient.

In this experiment, the archaeal compositions of R_ctrl_ and R_iron_ were more similar, and *Methanoculleus* was the predominant genus in the two samples (the relative abundance varied from 31% to 45%), followed by *Methanosarcina* (17–26%) and *Methanothrix* (19–27%), then *Methanomassiliicoccus* (9–11%), and other less abundant genera, such as *Methanobacterium*, *Methanospirillum*, *Methanosphaerula*, and *Methanoregula*. Palù M et al. found that adding *Methanoculleus bourgensis* into the mesophilic reactor improved the microbial metabolism and increased methane production by 11% [38]. It is worth noting that the proportion of hydrogenotrophic methanogens in R_iron_ increased by 15.4% compared to R_ctrl_. This shows that adding iron powder to the reactor can enhance the growth of hydrogenotrophic methanogens and increase the output of acetic acid and methane. This is an important reason why the methane generation of R_iron_ was more than R_ctrl_. Research has shown that adding iron powder to the reactor helps the microorganisms reduce obstacles to reproduction, and enhances metabolism at the same time. At the same time, it effectively increases the abundance of hydrogenotrophic methanogens, which helps reduce the hydrogen partial pressure in the reactor, thereby increasing acetic acid content and methane production [39].

The archaeal community composition in R_mix_ and R_high_ were similar. In both samples, *Methanoculleus* and *Methanosarcina* were the predominant genera, with relative abundance varying from 37% to 43% and 36% to 55%, respectively, followed by *Methanomassiliicoccus* (7–22%). Among them, hydrogenotrophic methanogens and mixed-nutrition methanogens that included hydrogen-consuming and acetate-consuming methanogens accounted for more than 99%, while acetoclastic methanogens were almost undetected. This may be because, after HTHP, the solubilization of organics in the sludge was increased, along with the content of short-chain fatty acids and ammonia nitrogen [21], affecting the microbial community in the reactors. One study showed that, in the process of AD, when the FAN concentration of the system was in the range of 250–500 mg·L^−1^, the microbial community was slightly inhibited, moderate inhibition occurred at 400–600 mg·L^−1^, and significant inhibition occurred with FAN concentrations of 600–800 mg·L^−1^ accompanied by a dramatic drop in biogas production [32]. Equation (2) may be used to calculate the FAN concentration in the reactors [40] where FAN is the concentration of free ammonia, TAN is the total ammonia concentration, and T is the temperature (kelvin). The TAN concentration in the R_mix_ and R_high_ were higher than R_ctrl_ during the reaction process, and as high as 2461.94 mg·L^−1^ and 2861.41 mg·L^−1^ at the end of the reaction (FAN concentration was 626.75 mg·L^−1^ and 589.30 mg·L^−1^). The archaeal community in the reactors were suppressed, but the biogas production exceeded that in R_ctrl_. This shows that HTHP screened out the archaeal community that can adapt to high FAN levels. However, it did not occupy a dominant position in the initial stage of anaerobic digestion, which is one of the reasons for the low methane production in the initial stage of the reactors. On the other hand, HTHP played a decisive role in the selection of the archaeal community in the R_mix_, but the increase in the abundance of the archaeal community was jointly determined by the HTHP and added iron powder.
(2)$$\rho \;(FAN)=\rho (TAN)\left[1+\frac{10^{-\mathrm{pH}}}{10^{-(0.09018+\frac{2729.92}{T})}}\right]$$

#### 3.2.3. Functional Abundance of Archaeal in Different Reactors

Figure 9 shows the functional abundance histogram. The higher the histogram, the greater the functional richness of the archaeal community. The total functional abundance of archaeal communities in the four reactors could be arranged in decreasing order as R_high_, R_mix_, R_iron_ and R_ctrl_, which was the same as the law of cumulative methane production. Among them, the order of carbohydrate transport and metabolism, as well as lipid transport and metabolism, could be arranged in decreasing order as R_iron_, R_mix_, R_ctrl_ and R_high_, while the order of amino acid transport and metabolism in decreasing order was R_high_, R_mix_, R_ctrl,_ and R_iron_. This shows that adding iron powder will reduce the carbohydrate transport and metabolism, as well as lipid transport and metabolism by archaeal community, but will increase the amino acid transport and metabolism. HTHP, however, had the opposite effect. Studies have shown that the functional abundance of amino acid metabolism, carbohydrate metabolism, and lipid metabolism can be improved, which is conducive to enhancing the degradation of macromolecular organics and increasing the cumulative methane production [41].

## 4. Conclusions

The results show that the three pretreatment methods successfully improved the methane production from anaerobic digestion, and could increase the peak value of methane production to varying degrees. They screened efficient archaeal communities, and even if the levels of VFA and TAN in the reactor were greatly improved. Their inhibiting effect on the methane production process was reduced. Among the four groups of reactors, the archaeal composition was most similar in R_ctrl_ and R_iron_, and the addition of iron powder promoted the growth of hydrogenotrophic methanogens. The archaeal composition in R_high_ and R_mix_ was more similar, and almost no acetoclastic methanogens were detected in the reactor. HTHP could screen the archaeal community that was suitable for high FAN levels. In the combined pretreatment method, HTHP played a decisive role in the boundary of the archaeal community, but the increase in the abundance of the archaeal was jointly determined by the HTHP and iron powder. At the same time, the addition of iron powder enhanced the transportation and metabolism of carbohydrates and lipids by archaeal community, while HTHP enhanced the amino acid transport and metabolism, which is an important reason for the increase in methane production.

## Figures and Tables

**Figure 1 ijerph-19-04470-f001:**
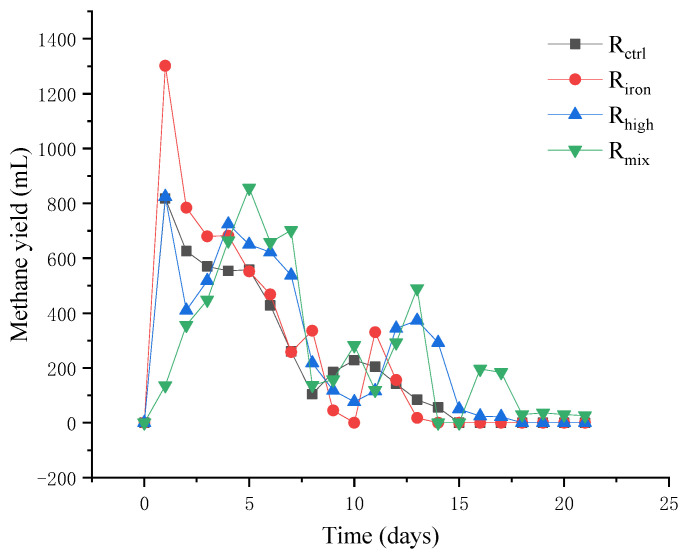
Methane yield of the four reactors.

**Figure 2 ijerph-19-04470-f002:**
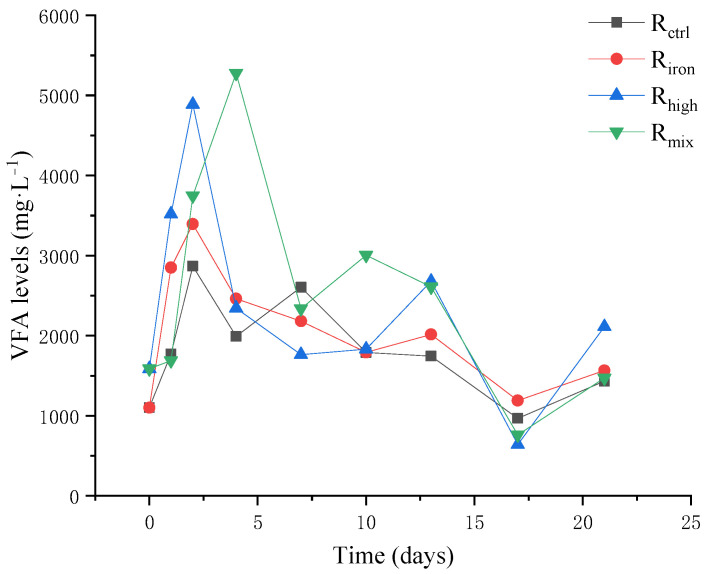
VFA levels of the four reactors.

**Figure 3 ijerph-19-04470-f003:**
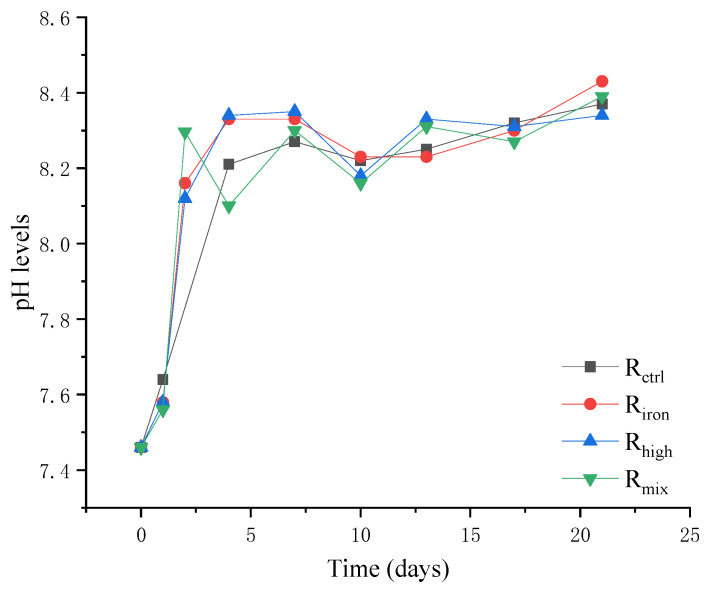
pH levels of the four reactors.

**Figure 4 ijerph-19-04470-f004:**
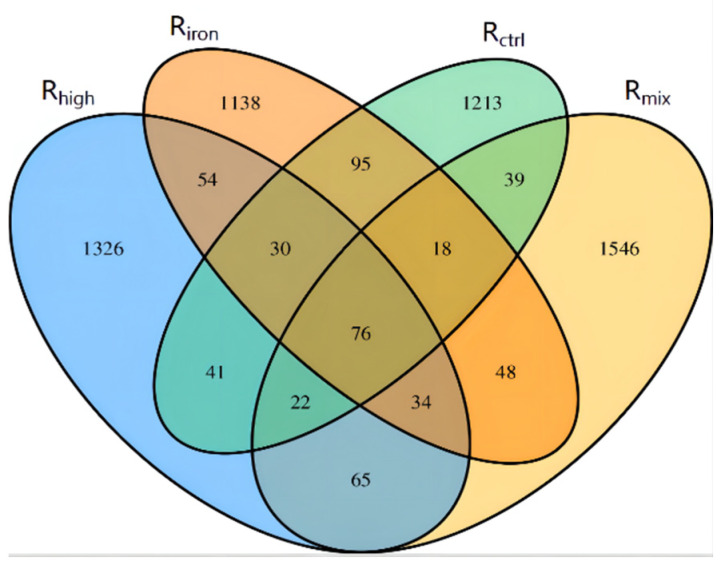
Venn diagram of the OTUs in the four samples. The numbers inside the diagram indicate the number of OTUs.

**Figure 5 ijerph-19-04470-f005:**
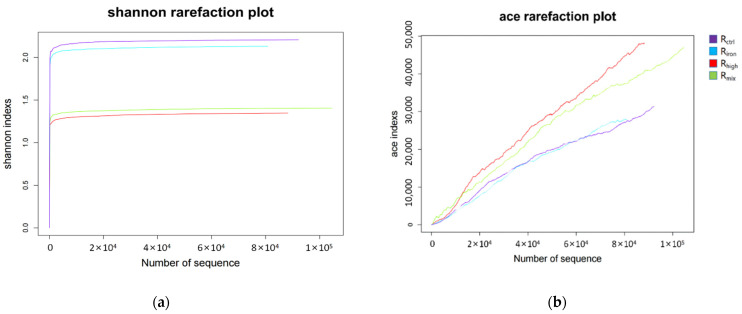
Alpha diversity based on the Shannon and Ace indexes at different experimental conditions. (**a**) Shannon rarefaction plot; (**b**) ace rarefaction plot.

**Figure 6 ijerph-19-04470-f006:**
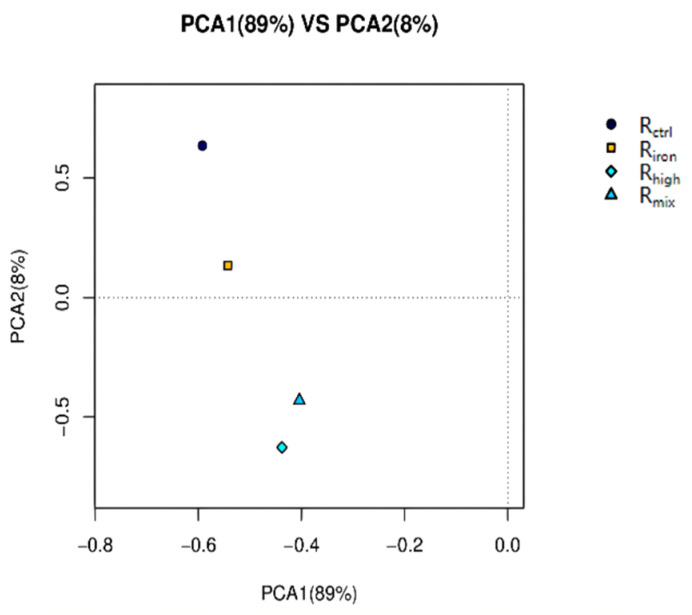
Principal component analysis from different reactors. PC1 and PC2 are the principal components 1 and 2, which represent 66.3% and 17.0% of community variation, respectively (based on weighted UniFrac metrics).

**Figure 7 ijerph-19-04470-f007:**
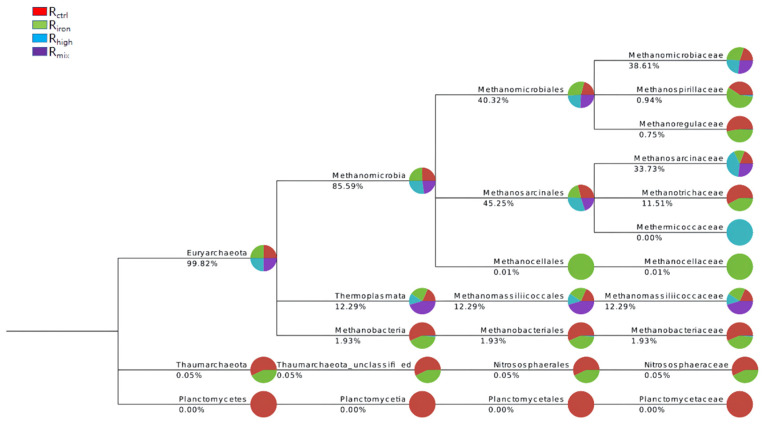
Changes in the composition of the archaeal community at different levels in the four reactors.

**Figure 8 ijerph-19-04470-f008:**
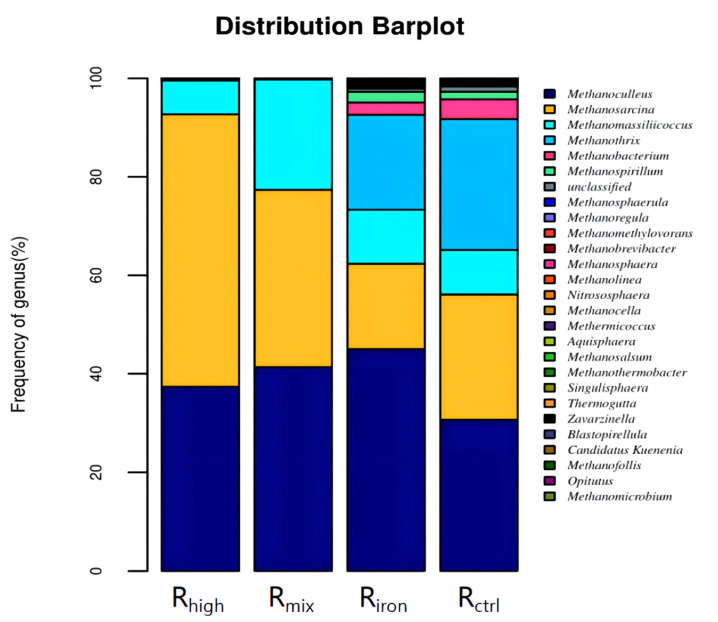
Archaeal communities at genus levels in four reactors.

**Figure 9 ijerph-19-04470-f009:**
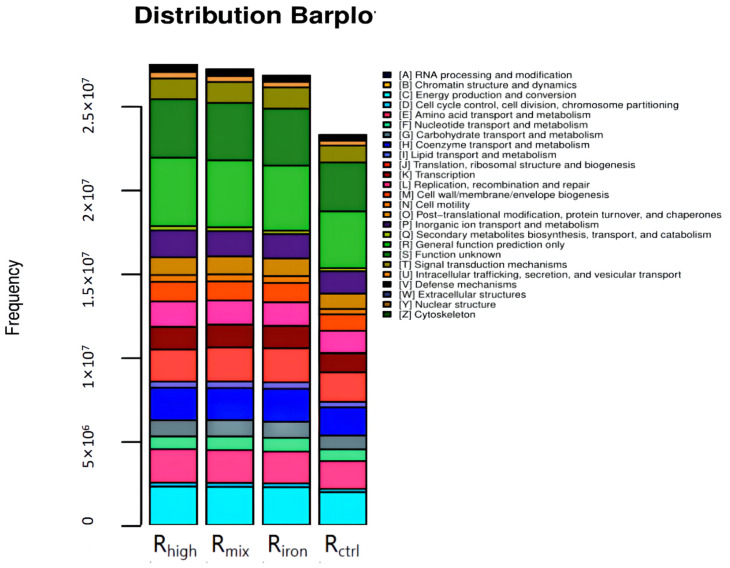
Functional abundance histogram of four reactors.

**Table 1 ijerph-19-04470-t001:** Physicochemical properties of sludge before anaerobic digestion.

Parameter	R_ctrl_	R_iron_	R_high_	R_mix_
TS (%)	6.73 ± 0.05	6.73 ± 0.05	7.00 ± 0.07	7.00 ± 0.07
VS (%)	3.77 ± 0.04	3.77 ± 0.04	3.75 ± 0.05	3.75 ± 0.05
SCOD (mg·L^−1^)	3616.00 ± 50.34	3616.00 ± 50.34	18,720.00 ± 100.89	18,720.00 ± 100.89
VFA (mg·L^−1^)	1101.30 ± 12.45	1101.30 ± 12.45	1586.76 ± 23.61	1586.76 ± 23.61
pH	7.46 ± 0.10	7.46 ± 0.10	7.46 ± 0.10	7.46 ± 0.10

**Table 2 ijerph-19-04470-t002:** Physicochemical properties of sludge after anaerobic digestion.

Parameter	R_ctrl_	R_iron_	R_high_	R_mix_
TS (%)	4.83 ± 0.01	5.91 ± 0.05	6.07 ± 0.03	6.09 ± 0.08
VS (%)	3.20 ± 0.01	3.07 ± 0.06	2.79 ± 0.02	2.85 ± 0.03
SCOD (mg·L^−1^)	5950.00 ± 84.48	7460.00 ± 56.63	14,120.00 ± 184.45	12,830.00 ± 147.86
VFA (mg·L^−1^)	1432.52 ± 18.56	1564.64 ± 20.84	2111.83 ± 15.74	1471.58 ± 121.87
pH	8.37 ± 0.12	8.43 ± 0.15	8.34 ± 0.10	8.39 ± 0.13

## Data Availability

The data that support the findings of this study are available from the corresponding author upon reasonable request.

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
