# Peer review of "Effects of Iron Powder Addition and Thermal Hydrolysis on Methane Production and the Archaeal Community during the Anaerobic Digestion of Sludge"

_ijerph, 2022, doi:10.3390/ijerph19084470_

Round 1
Reviewer 1 Report
The authors address a topic that is very important today. The treatment and recovery of wastewater and sludge is essential for the implementation of a circular economy.
The authors have done a thorough and careful job and I could not find any significant errors in the manuscript. Please find my observations and comments in the following lines:
- The Materials and Methods chapter does not discuss the mixing ratio of the two types of sludge, which can be a very important factor for reproducibility. I recommend that the authors make this correction.
-Explanations of abbreviations (e.g. TS, VS) are missing in the text. Although these are commonly used terms, their definition is necessary. Please review the text and add the missing abbreviation explanations.
-As before, Rctrl, Riron, Rmix and Rhigh are not defined, without them the text and figures are difficult to understand. I also recommend that the authors add these.
- I suggest increasing the resolution of Figures 4, 8 and 9 for better readability.
Author Response
Dear Editors and Reviewers:
Thank you for your letter and for the reviewers’ comments concerning our manuscript entitled “Effects of iron powder addition and thermal hydrolysis on methane production and archaeal community during anaerobic digestion of sludge” (ijerph-1626289). Those comments are all valuable and very helpful for revising and improving our paper, as well as the important guiding significance to our researches. We have studied comments carefully and have made correction which we hope meet with approval. Revised portion are marked in red in the text. The main corrections in the paper and the responds to the reviewer’s comments are as flowing:
- As for the mixing ratio of the two types of sludge, in order to simulate the operation state of the actual project, it is only necessary to adjust the total solids of the sludge to about 7%.
- I have added the explanation of abbreviations in the text.
- The explanation of the reactors (Rctrl, Riron, Rhigh and Rmix) has been added to the text.
- I have remade the Figures you mentioned.
We tried our best to improve the manuscript and made some changes in the manuscript. These changes will not influence the content and framework of the paper. And here we did not list the changes but marked in red in revised paper.
We appreciate for Reviewers’ warm work earnestly, and hope that the correction will meet with approval.
Once again, thank you very much for your comments and suggestions
Yours Sincerely
Yibin Wang
Reviewer 2 Report
What is the abbreviation of ‘THP’? You didn’t mention the abbreviation anytime in the paper.
What are the drawbacks of THP and other pretreatment methods? Why do we need new methods?
“However, research has not been undertaken that combines the two pretreatment methods to treat sludge.” Why is it important to combine these methods?
The significance of this study is not clear in the Introduction part. What is the rationale for doing this study?
According to Biological Methane Potential (BMP) method, storing digested sludge at 4°C is not ideal. It’s better to store them at the operating temperature at which the digester was run. Please, justify your storage method. I’ll suggest reviewing some papers on the BMP method.
Did you upload your sequencing data to any database? If not, please do that, and write the accession number.
What device did you use to measure the methane potential? How did you differentiate between CO2 and CH4 fractions? Have you done any GC analyses? The methodology of methane measurement is not clear.
The results in Fig. 1 are not consistent. Why there are such fluctuations in methane yield across different days? There should be a consistent pattern in gas production.
Please, re-draw Fig. 5 and Fig. 6. Fonts are too small to read properly.
Fonts in Fig. 7 are very difficult to follow.
Please re-check the writing. In many places, there are no spaces between two words.
Author Response
Dear Editors and Reviewers:
Thank you for your letter and for the reviewers’ comments concerning our manuscript entitled “Effects of iron powder addition and thermal hydrolysis on methane production and archaeal community during anaerobic digestion of sludge” (ijerph-1626289). Those comments are all valuable and very helpful for revising and improving our paper, as well as the important guiding significance to our researches. We have studied comments carefully and have made correction which we hope meet with approval. Revised portion are marked in red in the text. The main corrections in the paper and the responds to the reviewer’s comments are as flowing:
- The abbreviation of THP is Thermal hydrolysis pretreatment, I have added in the text.
- The disadvantage of THP is that it has high requirements for equipment,adding iron powder and other methods may have a bad impact on sludge dewatering and sludge disposal. Therefore, in order to facilitate the subsequent sludge treatment, at the same time, increase methane production and help sewage treatment plants achieve carbon neutralization. we need to explore new methods.
- The reason for combining these methods is thermal hydrolysis pretreatment enhanced the anaerobic digestion methane production process from the hydrolysis stage. The addition of iron powder strengthened the acidogenesis, acetogenesis, and methanogenesis stages. Therefore, it is considered to combine the two pretreatment methods to strengthen the anaerobic digestion methane production process in four stages.
- I think there are two reasons for this study: one is to explore the impact of different pretreatment methods on methane production, and the other is the impact of different pretreatment methods on the Archaeal community in the reactor.
- digested sludge is not stored at 4°C, we used it immediately after taken from the anaerobic digestion tank. Dewatered sludge stored in a refrigerator at 4°C before use. This is a mistake in my writing.
- I'm very sorry about the sequencing data. I have uploaded it as a supplementary material.
- I use NaOH solution to absorb CO2 in gas production and the content of methane is calculated according to the volume of NaOH solution discharged. Due to the limitation of test instruments, I have not done GC analyses.
- Same pretreatment method should be a consistent pattern in gas production. However, different pretreatment methods change the internal environment of the reactors. Microbial community, physical and chemical indicators, these are the reasons for the fluctuations of methane production.
- I have remade the Figures you mentioned.
- I'm very sorry about the writing. I've rechecked the problem of spaces.
We tried our best to improve the manuscript and made some changes in the manuscript. These changes will not influence the content and framework of the paper. And here we did not list the changes but marked in red in revised paper.
We appreciate for Reviewers’ warm work earnestly, and hope that the correction will meet with approval.
Once again, thank you very much for your comments and suggestions
Yours Sincerely
Yibin Wang